# Effects of Substrate and Cement Shade on the Translucency and Color of CAD/CAM Lithium-Disilicate and Zirconia Ceramic Materials

**DOI:** 10.3390/polym14091778

**Published:** 2022-04-27

**Authors:** Allegra Comba, Gaetano Paolone, Andrea Baldi, Alessandro Vichi, Cecilia Goracci, Giulia Bertozzi, Nicola Scotti

**Affiliations:** 1Department of Surgical Sciences, Dental School Lingotto, University of Turin, 10126 Turin, Italy; allegra.comba@unito.it (A.C.); andrea.baldi@unito.it (A.B.); giulia.biri@hotmail.it (G.B.); 2Department of Dentistry, IRCCS San Raffaele Hospital and Dental School, Vita Salute University, 20132 Milan, Italy; paolone.gaetano@hsr.it; 3Dental Academy, Faculty of Science and Health, University of Portsmouth, William Beatty Building, Hampshire Terrace, Portsmouth PO1 2QG, UK; alessandro.vichi@port.ac.uk; 4Department of Medical Biotechnologies, University of Siena, Policlinico Le Scotte, 53100 Siena, Italy; cecilia.goracci@unisi.it

**Keywords:** translucency, color stability, cement, zirconia, lithium disilicate, ceramic, opacity

## Abstract

The aim of this in vitro study was to evaluate the effects of substrate and cement shades on the translucency and color of lithium-disilicate and zirconia CAD/CAM materials. Two light-cured resin cements (RelyX Veneer Cement; 3M; Choice 2 Veneer Cement; Bisco Dental) with a standardized thickness (0.1 mm) were tested in combination with two different monolithic CAD/CAM materials (E-Max CAD (LI_2_SI_2_O_5_); Ivoclar Vivadent; Katana (ZrO_2_); Kuraray-Noritake Dental) on two different colored composite substrates used as a dentin (Filtek Supreme XTE; 3M); for a total of 12 combinations (*n* = 10). The specimens’ color was measured with a spectrophotometer (Spectroshade; MHT). Measurements were taken using the CIELAB color coordinate system (*L***a*b**) against black and white backgrounds. *L***a*b** values were statistically analyzed for the variables Substrate, Ceramic, and Cement by applying a Three-Way ANOVA followed by the Tukey Test for *post-hoc* comparison (*p* < 0.05). Translucency Parameter (TP) and Constant Ratio (CR) were assessed to evaluate translucency; acceptability and perceptibility thresholds (Δ*E*_00_ 1.8 and 0.8) were used. Statistically significant influence was found for factors ceramic material, cement shade, and substrate color (*p* < 0.05). Unacceptable color differences were reported for Li_2_Si_2_O_5_. Opacity was significantly higher when white opaque cement shade was employed. Ceramic type and cement shade significantly influenced *L***a*b** color coordinates. The final translucency and color of ceramic restorations can, therefore, be influenced by ceramic material, cement shade, and substrate color.

## 1. Introduction

In clinical situations involving esthetic restorations, ceramic materials have the capability to replicate the appearance and the optical properties of natural teeth and are, therefore, suitable for manufacturing several types of all-ceramic restorations for anterior and posterior teeth, such as inlays, onlays, crowns, veneers, and bridges [1]. Contemporary all-ceramic materials in conjunction with adhesive systems and luting cements allow clinicians to use a minimally invasive approach and to make more conservative restorations obtaining excellent esthetic results.

Since the introduction of all-ceramic materials, the limitation in light transmission typical of porcelain-fused to metal (PFM) restorations has been overcome. Furthermore, the introduction of CAD/CAM technology facilitated the spread of these newer materials providing the clinician several opportunities to treat esthetic challenging cases. In PFM restorations, the metal was covered by a white and opaque material that represented a standardized substrate. In all-ceramic materials, substrate and cement color are variable, as well as the different levels of translucency of the ceramic itself [2]. Despite the unquestionable better esthetic outcome, all-ceramic materials introduced a higher level of complexity to the shade matching process. The final outcome of all-ceramic restorations is not limited to shade selection, but rather the resulting of several factors, such as the translucency degree; the restoration thickness; the surface properties, such as roughness and gloss; the shade and the opacity of the luting agents; and the substrate [3,4,5,6]. Ceramic materials with high translucencies allow more light to enter and scatter; consequently, the underlying tooth shade could have a significant influence over the final color.

The influence of the above-mentioned factors on translucency has been reported for felspathic ceramic [7], lithium-disilicate [8,9], and zirconia materials [10,11]. However, few studies and limited clinical guides correlating the luting cement and the background substrate shade on translucency and color of lithium disilicate compared to last generation cubic zirconia are present in the literature. For this reason, the influence of resin cement and substrate shade on translucency and color was investigated in this study.

The formulated null-hypotheses were:After the cementation procedure, resin cement does not influence the final shade of lithium disilicate and cubic zirconia restorations;Substrate color does not influence translucency and color of lithium disilicate and cubic zirconia restorations.

## 2. Materials and Methods

### 2.1. Study Design

The general description of the main materials used in the present study, their manufacturers and composition are listed in Table 1.

This study was designed in 12 study groups (*n* = 10 each), where the specimens were randomly allocated (www.randomizer.org, access on 16 November 2019) considering:“CAD/CAM ceramic material” in 2 levels: lithium disilicate (E-Max CAD, Ivoclar Vivadent, Schaan, Liechtenstein) and tetragonal zirconia discs (Katana, Kuraray Noritake Dental, Tokyo, Japan) were used to simulate indirect anterior restorations;“Luting cement shade” in 3 levels: two completely different cement shades (Translucent Rely-X Veneer, 3M; Milky Opaque, Choice 2, Bisco, Chicago, IL, USA) and a transparent glycerin gel as negative control were used to understand the effect of different shades on ceramic optical properties;“Background shade” in 2 levels: to simulate different shades of prepared teeth, two composite shades, A2 and A4, were used to create disk-shaped samples.

### 2.2. Sample Preparation

A total of 120 disks (10 mm diameter × 2 mm thickness) of Filtek Supreme XTE, (3M ESPE, St. Paul, MN, USA) of two different shades (A2D and A4D, *n* = 60), were manufactured. A calibrated mold was employed to control the diameter and thickness of the disks. A glass plate was pressed on the surface of the mold to eliminate the excess of material and then polymerization was performed by a LED curing unit (Valo, Ultradent Products, South Jordan, UT, USA) for 20 s with a light intensity of 1200 mW/cm^2^. The specimens were then immersed in distilled water for 24 h to allow completion of polymerization. Then, 60 CAD/CAM lithium-disilicate (Li_2_Si_2_O_5_) blocks (E-Max CAD, Ivoclar Vivadent, Schaan, Liechtenstein) were cut perpendicularly with a water-cooled low speed diamond saw (Isomet, Buehler, Lake Bluff, IL, USA). Flat 14 mm × 12 mm specimens of 1.2 mm thickness were cut. The block was maintained perpendicular to the saw during cutting, to ensure a consistent thickness of specimen. After sectioning, all samples were polished and crystallized according to the manufacturer instructions.

I total of 60 Flat 14 mm × 12 mm specimens of 1.2 mm thickness were designed and milled from CAD-CAM tetragonal zirconia discs (ZrO_2_) (Katana, Kuraray Noritake Dental, Tokyo, Japan). After milling specimens were sintered, finished, and glazed according to the manufacturer instructions.

Specimens were then randomly divided into three subgroups (*n* = 20 each, 10 per ceramic material) according to the luting cement shade: Translucent (TR) (RelyX veneer, 3M ESPE, St. Paul, MN, USA); Milky Bright (MB) (Choice 2, Bisco Dental, Schaumburg, IL, USA); and glycerine transparent gel (negative control).

After sandblasting of composite and ceramic specimen (50 μm, 2 atm, 2 cm, 20 s), ceramic disks were then luted on composite disks with simplified adhesive procedures, which were conducted as follows: on the composite, silane was applied (Bis-Silane, Bisco Dental, Schaumburg, IL, USA) and, after evaporation with air, a universal adhesive system was used following the manufacturer instructions (All Bond Universal, Bisco Dental, Schaumburg, IL, USA). On lithium disilicate, 4% hydrofluoric acid (Bisco Porcelain Etchant Gel, Bisco Dental, Schaumburg, IL, USA) was applied for 20 s and then rinsed for 10 s. Disks were immersed in an ultrasonic bath with 90% ethanol for 5 min; after drying in air, silane, and universal adhesive were applied to the specimens following manufacturer instructions.

On ZrO_2_ specimen, 10-MDP primer was applied (Z-Prime, Bisco Porcelain Etchant Gel) and, after evaporation with air, the universal adhesive system was applied as previously described. To standardize the luting cement thickness, a metal plate 0.1 mm thick with 9 mm diameter holes was employed. Composite disks, previously treated with adhesion techniques described above, were placed below the metal plate, in correspondence of the holes. A drop of cement was applied on the composite surface inside the hole of the metal plate, and on the other side of the metal mold the ceramic disk was placed. A standardized seating force of 40 g/mm^2^ (equivalent to a force of about 70 N, which can be considered a medium seating force [12,13] was applied on the ceramic specimens.

After performing the pressure to remove the excess of cement, curing was carried out on both sides for 60 s with the above-mentioned LED curing light unit. A schematic representation of a sample preparation is represented in Figure 1.

### 2.3. Constant Ratio (CR), Translucency Parameter (TP), and Color Difference Measurements

After 24 h storage in distilled water, color coordinates were assessed using a spectrophotometer (Spectroshade, MHT, Niederhasli, Switzerland) applied on the ceramic side. Measurements were taken using the CIELAB color coordinate system (*L***a*b**) against black (CIELAB 0,0,0–3% reflectivity) and white (CIELAB 100,0,0–90% reflectivity) backgrounds (Kodak Gray Scale Q-14, Rochester, NY, USA).

The spectrophotometer displayed an image on the screen: an operator delimited three separate internal reading areas (3 mm diameter each) near the center of the specimen, in order not to be influenced by the metal plate. The mean value of the three measurements was obtained.

*L***a*b** values were statistically analyzed for the variables Substrate, Ceramic, and Cement by applying a Three-Way ANOVA followed by the Tukey Test for *post-hoc* comparison (*p* < 0.05).

The TP was calculated using the following CIELAB color difference equation:$$\mathrm{TP}=\sqrt{{\left(L_{B}^{*}-L_{w}^{*}\right)}^{2\;}+{\left(a_{B}^{*}-a_{w}^{*}\right)}^{2}+{\left(b_{B}^{*}-b_{w}^{*}\right)}^{2}}$$
where the subscripts “*b*” and “*w*” refer to color coordinates over the above-mentioned black and white backgrounds.

CR is the ratio between the reflectance of a specimen on a black background to that on a white background of a known reflectance [14]. The CR values are calculated according to the equation:$$\mathrm{CR}=\frac{y_{B}}{y_{W}}$$
where *y_B_* represents the spectral reflectance of the light of the specimen on a black background and *y_W_* on a white background.

*y* is calculated with the equation:$$y={\left[\frac{\left(L+16\right)}{116}\right]}^{3}\times 100$$

The CR is a direct measure of opacity, and it decreases as translucency increases. The value of a perfectly transparent material is 0, while the value of a completely opaque material is 1 [15]. To calculate differences in color between ZrO_2_/Li_2_Si_2_O_5_ and control group, the CIEDE2000 formula was applied.
$$\Delta E_{00}=\sqrt[{2}]{{\left(\frac{\Delta L^{\prime }}{K_{L}S_{L}}\right)}^{2}+{\left(\frac{\Delta C^{\prime }}{K_{C}S_{C}}\right)}^{2}+{\left(\frac{\Delta H^{\prime }}{K_{H}S_{H}}\right)}^{2}+R_{T}\left(\frac{\Delta C^{\prime }}{K_{C}S_{C}}\right)\left(\frac{\Delta H^{\prime }}{K_{H}S_{H}}\right)}$$
where Δ*L*′, Δ*C*′, and Δ*H*′ are the differences in lightness, chroma, and hue, respectively, and R_T_ is a function that accounts for the interaction between chroma and hue differences in the blue region. Weighting functions, *S_L_, S_C_,* and *S_H_* adjust the total color difference for variation in the location of the color difference pair in *L*′, *a*′, and *b*′ coordinates and the parametric factors, *K_L_*, *K_C_*, and *K_H_*, are correction terms for experimental conditions. In the present study, the parametric factors of the CIEDE2000 color difference formula were set to 1 [16,17,18].

The acceptability (AT) and perceptibility (PT) thresholds were set, respectively, at Δ*E*_00_ 1.8 and 0.8 [17,18,19].

### 2.4. Statistical Analysis

Normal distribution of color coordinates was verified with Shapiro–Wilk test. To evaluate the effect of the factors “ceramic material” (Li_2_Si_2_O_5_ or ZrO_2_), “luting cement” (TR, MB), and “background shade” (A2D, A4D) on color coordinates and translucency, a three-way ANOVA test was performed for CR, TP, and *L***a*b** variables. Post hoc pairwise comparison was performed using Tukey test. The significance level was set at 95% (*p* < 0.05). All statistical analyses were performed using the Stata 14.1 software package (StataCorp, 4905 Lakeway Drive, College Station, TX 77845, USA).

## 3. Results

The mean values and standard deviation for CR and TP obtained in the different groups were reported in Table 2 and Table 3, respectively. A three-way ANOVA test showed that CR was significantly influenced by “background shade”, “ceramic material”, and “luting cement”. Considering CR, a post hoc Tukey test showed that among cements, the opacity was significantly higher when MB cement shade was employed. No differences were reported between TR and control group (glycerin). Regarding the ceramic material, ZrO_2_ significantly affected the CR more than Li_2_Si_2_O_5_, while among background shade, the CR was higher when A2 was employed as background.

TP was significantly affected by “background shade” and “ceramic material”, but no significant differences were observed for the factor “luting cement”. Higher TP was observed when A2 background was used. ZrO_2_ provided significantly less TP values in respect to Li_2_Si_2_O_5_.

Regarding *L***a*b** values, means and standard deviation obtained in different groups were displayed in Table 4 and Table 5. The *L**** coordinate, that represents value/luminosity, was significantly affected by background shade, ceramic material, and luting cement. A *post-hoc* Tukey test showed among cement significant differences were reported between both cements and control group. Coordinate *a**, that is relative to the green-red opponent colors, was affected by ceramic material and luting cement, but not by the background shade. No significant differences were observed between TR cement and control group. Coordinate *b**, that is relative to the blue-yellow axis, was significantly influenced by ceramic and cement, but not by background.

## 4. Discussion

The present study evaluated the influence of background substrate and luting cement shades on the translucency and color of lithium disilicate and cubic zirconia manufactured using CAD/CAM technology. Based on the obtained results, the background substrate and the luting cement shades significantly affected the translucency and final color of evaluated ceramic materials. Therefore, both study hypotheses were rejected.

Through a spectrophotometer, CIELab color coordinates were obtained and used to calculate TP and CR for translucency analysis. In previous studies, CIEDE2000 formula, which was also employed in the present study to determine color differences, showed better results related to color differences perceived by human eye than the CIELab formula [17,18,20,21] Color differences values were compared with the visual threshold Li_2_Si_2_O_5_ (PT and AT) reported for the used metric [17,22,23,24]. The present study findings suggested that the color differences observed were clinically acceptable for ZrO_2_ and completely unacceptable for Li_2_Si_2_O_5_. The reason of these difference can be advocated to the relationships between microstructure, composition, manufacturing process, and properties of dental ceramics [25]. Previous studies showed how optical and mechanical properties of ceramics essentially depend on type, shape, size of particles, refractive index (RI), and on the distribution of the crystalline phase [26,27,28,29].

Ceramic materials can be composed of particles bigger or smaller than visible light wavelength (that ranges from 0.4–0.7 μm) [29]. Refraction and reflection occur on the surfaces of particles larger than the wavelength of light. The greater the RI difference between particle size and matrix, the greater the refraction and reflection of light, which leads to higher opacity [30]. The differences in color and translucency of the investigated CAD/CAM ceramics may be due to the crystal structure, in particular the crystal size and the relationship between glass phase and crystals, which could strongly affect the light refraction and transmission [31]. Translucency of cubic zirconia has been investigated extensively in the recent literature and it is strictly related to the proportion of the crystalline phases, being lower with the tetragonal and higher with the cubic one [32,33,34]. Wang et al. [7] reported lower translucency for monolithic zirconia (ranging from 5.5–13.5) in respect to human dentin (16.4) and enamel (18.1). Comparing the two tested materials, higher translucency was reported for Li_2_Si_2_O_5_. This difference can be advocated to the combination of a glass matrix and crystalline phase, which could reduce internal scattering of the light as it passes through the lithium disilicate material [35,36]. Moreover, the main crystalline phase is characterized by elongated lithium disilicate (Li_2_Si_2_O_5_) crystals that represent a randomly oriented scaffold with a length ranging from 3–6 μm. The second phase is composed by a lower volume of lithium orthophosphate (Li_3_PO_4_), which is responsible for the translucency of the material [37]. Li_2_Si_2_O_5_ restorations may, therefore, be influenced by ceramic thickness, translucency, substrate color, and cement color as reported by several authors [36,38,39,40,41,42]. The difference between ZrO_2_ and Li_2_Si_2_O_5_ highlighted in the present study are consistent with a recent study by Church et al., which reported that even the most translucent zirconia is not as translucent as lithium disilicate [3].

When considering translucency and color, the thickness is another important factor with materials such as resin composite and ceramic. Increasing thickness will help the masking capability, thus improving the overall shade. As the ceramic thickness increases, the diffused reflection mainly occurs in the ceramic itself than from the underlying substrate. [39,40,43,44]. This effect is implemented by the intrinsic translucency of the material [8]. In addition to the translucency variation, the material’s thickness may have consequences on light irradiation intensity through the material, which could, therefore, affect the conversion degree of the underlining luting cement [8,34,36]. A recent study by Tafur Zelada et al. [45] pointed out that the decrease of light irradiation intensity through thick zirconia frameworks could lead to color instability and poor mechanical properties.

Translucency is not only influenced by ceramic thickness or chromaticity, but also by other factors, such as the luting cement shade [3,38,46,47,48]. In the present study, two cement colors were evaluated, with the white opaque one (MB) showing significantly more influence on translucency and color over the translucent (TR) one. These findings were in accordance with Chaiyabutr et al. [38] who reported lower ∆E values in the group with opaque cement color. On the other hand, Guazzato et al. [49] reported that cement color has no influence on final color, but these inconsistencies can be related to the different thicknesses of the cement layer that were simulated in different studies.

The opaque cement resulted in lower color change, thus providing better substrate masking ability [2,50,51,52], and it could, therefore, provide color correction in clinical situation depending on ceramic material translucency [53]. The capability of opaque cements to influence color on all-ceramic restorations may be advocated to cement composition [54]. The inorganic fillers that provide opacity are characterized by different refractive index, with subsequent scattering of light and different degrees of translucency [55]. It also must be considered that cement tint saturation influences the translucency level, therefore influencing final color as well: lower chroma shades are in fact more translucent than saturated ones [56].

Another important aspect to consider in color and translucency studies is whether the underlying substructure can affect the final color of an esthetic restoration. The obtained results showed that background shade significantly influenced translucency and color of tested ceramic materials. Moreover, the masking ability of white opaque cements is limited, and, consequently, it can be stated that the final color of high-translucency Li_2_Si_2_O_5_ and ZrO_2_ restorations is mainly affected by the core material shade. This finding is in accordance with previous studies, which generally highlighted how a dark background, such as a dyschromic tooth or a metallic post-and-core build-up, could be difficult to mask with metal-free restorations [38,39,40,42,43,48]. An interesting study by Dotto et al. [57] reported that changing substrate color by layering lighter opaque composites on the abutment was able to provide lower ΔE values, thus influencing positively color match.

Among the limitation of present study, the variables associated with aging-induced color changes could be mentioned. Zirconia may in fact show a change in translucency after aging, which could be attributed to tetragonal-to-monoclinic phase transformation [58]. Aging could also affect cement color stability, eventually shifting to yellow due to water absorption by components such as triethyleneglycol dimethacrylate and 2,2-bis (4-[2-hydroxy-3-methacryloyloxy] phenyl) propane. Color may also be influenced in time from uncured camphorquinone depending on the polymerization rate that, as previously mentioned, can be affected by thickness and opacity of the ceramic material [42].

## 5. Conclusions

Based on the results and within the limitation of the present study, it can be concluded that the final shade and translucency of ceramic restorations can be significantly influenced by the type of material, cement, and substrate.

Opaque cements could be useful in influencing the color of the restoration in cases of a color mismatch within the desired shade due to the restoration’s shade or substrate color.

## Figures and Tables

**Figure 1 polymers-14-01778-f001:**
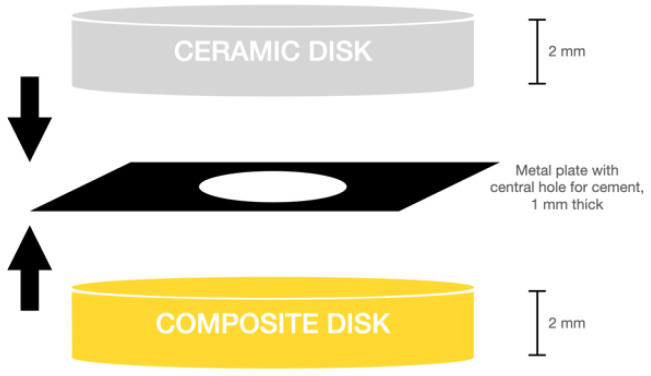
Sample preparation sketch.

**Table 1 polymers-14-01778-t001:** Materials used in present study.

Abbreviation	Name	Shade	Manufacturer	Type of Material
LI_2_SI_2_O_5_	E-Max CAD	A2 LT	Ivoclar Vivadent, Schaan, Liechtenstein	Lithium disilicate glass ceramic
ZrO_2_	Katana	HT12	Kuraray Noritake Dental, Tokyo, Japan	3Y-TPZ Zirconia
A2, A4	Filtek Supreme XTE	A2D, A4D	3M ESPE, St. Paul, MN, USA	Nanofilled resin-based composite
TR	RelyX Veneer Cement	TR	3M ESPE, St. Paul, MN, USA	Light cured cement
MB	Choice 2 Veneer Cement	Milky Bright	Bisco Dental, Schaumburg, IL, USA	Light cured cement
n.a.	Bis-Silane	n.a.	Bisco Dental, Schaumburg, IL, USA	Coupling agent
n.a.	All Bond Universal Adhesive	n.a.	Bisco Dental, Schaumburg, IL, USA	Universal adhesive
n.a.	Bisco Porcelain Etchant Gel	n.a.	Bisco Dental, Schaumburg, IL, USA	Hydrofluoric acid
n.a.	Z-Prime Plus Primer	n.a.	Bisco Dental, Schaumburg, IL, USA	Zirconia primer

**Table 2 polymers-14-01778-t002:** Means and standard deviation for CR. Different superscript capital letters indicate significant difference within column (*p* < 0.05). Different superscript lower-case letters indicate significant difference within rows (*p* < 0.05).

	Background A2			Background A4		
Cement Shade	Control	MB	TR	Control	MB	TR
Li_2_Si_2_O_5_	0.12 ± 0.01 ^B,bc^	0.17 ± 0.02 ^B,a^	0.17 ± 0.01 ^B,a^	0.10 ± 0.01 ^B,c^	0.14 ± 0.02 ^B,ab^	0.12 ± 0.01 ^B,bc^
ZrO_2_	0.45 ± 0.04 ^A,ab^	0.45 ± 0.02 ^A,a^	0.40 ± 0.03 ^A,b^	0.42 ± 0.04 ^A,ab^	0.41 ± 0.02 ^A,b^	0.42 ± 0.03 ^A,ab^

**Table 3 polymers-14-01778-t003:** Means and standard deviation for TP. Different superscript capital letters indicate significant difference within column (*p* < 0.05). Different superscript lower-case letters indicate significant difference within rows (*p* < 0.05).

Background	A2			A4		
Cement Shade	Control	MB	TR	Control	MB	TR
Li_2_Si_2_O_5_	2.86 ± 1.33 ^A,a^	1.18 ± 0.47 ^A,b^	3.61 ± 0.60 ^A,a^	1.90 ± 0.55 ^A,b^	1.75 ± 0.98 ^A,b^	1.58 ± 0.58 ^A,b^
ZrO_2_	0.69 ± 0.29 ^B,a^	0.71 ± 0.23 ^A,a^	0.84 ± 0.37 ^B,a^	0.51 ± 0.23 ^B,a^	0.63 ± 0.28 ^B,a^	0.56 ± 0.31 ^B,a^

**Table 4 polymers-14-01778-t004:** Means and standard deviation for *L***a*b** values. Similar superscript letters indicate no statistically significant difference (*p* > 0.05). Different superscript capital letters indicate significant difference within column (*p* < 0.05).

Background	Ceramic	Cement	*L**	*a**	*b**
A2	Li_2_Si_2_O_5_	Control	65.4 ± 1.19 ^D^	−0.46 ± 0.27 ^BC^	20 ± 1.37 ^AB^
		MB	69.84 ± 1.82 ^C^	−0.58 ± 0.39 ^BC^	13.3 ± 2.05 ^C^
		TR	70.6 ± 1.07 ^C^	−0.74 ± 0.62 ^B^	13.52 ± 1.92 ^C^
	ZrO_2_	Control	85.4 ± 1.41 ^A^	−0.64 ± 1.18 ^C^	1.34 ± 0.39 ^D^
		MB	85.7 ± 0.83 ^A^	−0.9 ± 0.18 ^A^	0.98 ± 0.57 ^D^
		TR	83.38 ± 1.36 ^B^	−1.02 ± 0.1 ^A^	0.64 ± 0.44 ^D^
A4	Li_2_Si_2_O_5_	Control	62.9 ± 0.56 ^E^	−0.46 ± 0.17 ^BC^	18.58 ± 1.11 ^B^
		MB	67.94 ± 1.76 ^C^	−0.68 ± 0.43 ^C^	20.68 ± 1.25 ^A^
		TR	65.88 ± 0.7 ^D^	−0.3 ± 0.27 ^BC^	11.96 ± 1.01 ^C^
	ZrO_2_	Control	84.3 ± 1.73 ^AB^	−0.56 ± 0.17 ^C^	1.32 ± 0.26 ^D^
		MB	84.08 ± 0.81 ^AB^	−0.72 ± 0.28 ^C^	0.82 ± 0.36 ^D^
		TR	84.4 ± 1.09 ^AB^	−0.78 ± 0.23 ^A^	0.68 ± 0.51 ^D^

**Table 5 polymers-14-01778-t005:** ⊗*E*_00_ values of MB and TR vs. control group. Values above acceptability (1.8) and perceptibility (0.8) threshold are outlined (* ^#^).

Background	A2		A4	
Cement Shade	MB	TR	MB	TR
Li_2_Si_2_O_5_	5.22 *	5.55 *	4.28 *	4.66 *
ZrO_2_	0.55	1.60 ^#^	0.5	0.70

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
