# Peer review of "Effects of Substrate and Cement Shade on the Translucency and Color of CAD/CAM Lithium-Disilicate and Zirconia Ceramic Materials"

_polymers, 2022, doi:10.3390/polym14091778_

Round 1
Reviewer 1 Report
Dear authors,
please find my comments and corrections on your text:
-lithium-disilicate without a dash, is a compound (Li2Si2O5), not a composite material. No use of (LD) as abbreviation, it is not a brand, the formula is the right representation of the compound. Correct throughout the text.
-zirconia is the ZrO2. if you mean that use the formula instead of ZR in text. if you mean other oxides or silicates based on zirconia or mixed with zirconia, please specify. Check and correct throughout the text.
-L*a*b*: correct to L*, a*, b* correct throughout the text (like x, y, z)
-latin phrases in italics: in vitro, post-hoc etc
-ΔE00: 00 as subscripts? Do you mean ΔΕ* or sth else? Correct throughout the text.
-Are you sure the citations are in parenthesis and not brackets [1, 2]?
-explain a bit "felspathic ceramic"
-l. 76: a space line after table. Correct throughout the text
-1200 mW/cm2: always a space between the number and the unit, correct where needed.
-5 min, instead of minutes. "Rinsed" with deionized H2O I suppose?
-you have incorporated too many specimens, referring to materials and conditions applied. Consider sketching a chart on the separation of the specimens and the treatment each group has suffered, for Experimental part.
-l. 132: b or B? l. 137: Yb? eq. 3: L*?
-l. 144: "ΔC0, and ΔH0" are mentioned? Check the typing of variables in l. 144-149.
-Table 2, 3: reduce font size to have superscripts in same raw. The decimal symbol in English is the dot.
-'L': no need for ', likewise a, b*
-"process, and": never use a comma before "and" or "or" when for parathesis of similar things.
-references: same formatting in all section
You have tried to attribute the colour changes in the crystalline microstructure correctly, what about the influence of HF solution or the etching technique? What is your proposition? What about the human-eye capability of identifying the colour changes (ΔΕ*>5)? What about restorations in front teeth (vertical reflexion) or the back teeth (horizontal replacements)? Please comment and correlate your results where possible.
Author Response
-lithium-disilicate without a dash, is a compound (Li2Si2O5), not a composite material. No use of (LD) as abbreviation, it is not a brand, the formula is the right representation of the compound. Correct throughout the text.
Our response: thanks for the suggestion, the text was accordingly modified
-zirconia is the ZrO2. if you mean that use the formula instead of ZR in text. if you mean other oxides or silicates based on zirconia or mixed with zirconia, please specify. Check and correct throughout the text.
Our response: thanks for the suggestion, the text was accordingly modified
-L*a*b*: correct to L*, a*, b* correct throughout the text (like x, y, z)
Our response: thanks for the suggestion, the text was accordingly modified
-latin phrases in italics: in vitro, post-hoc etc
Our response: thanks for the suggestion, the text was accordingly modified
-ΔE00: 00 as subscripts? Do you mean ΔΕ* or sth else? Correct throughout the text.
Our response: thanks for the suggestion, the text was accordingly modified
-Are you sure the citations are in parenthesis and not brackets [1, 2]?
Our response: thanks for the suggestion, the text was accordingly modified
-explain a bit "felspathic ceramic"
Our response: thanks for the question. Feldspatheic porcelain is a highly translucent, esthetic material for restorations fabricated with the traditional veneering porcelain powder and liquid brush buildup technique. It is just mentioned in the introduction since previous studies investigated the color of this material, but it is not a CAD/CAM material and it has huge differences to lithium disilicate and zirconia.
-l. 76: a space line after table. Correct throughout the text
Our response: thanks for the suggestion, the text was accordingly modified
-1200 mW/cm2: always a space between the number and the unit, correct where needed.
Our response: thanks for the suggestion, the text was accordingly modified
-5 min, instead of minutes. "Rinsed" with deionized H2O I suppose?
Our response: thanks for the comment. In Dental papers, “rinse” refers with H2O without being deionized
-you have incorporated too many specimens, referring to materials and conditions applied. Consider sketching a chart on the separation of the specimens and the treatment each group has suffered, for Experimental part.
Our response: thanks for the suggestion, the text was accordingly modified and a schematic representation of a sample preparation was added.
-l. 132: b or B? l. 137: Yb? eq. 3: L*?
Our response: thanks for the comment, the formula is correct as it is already written
-l. 144: "ΔC0, and ΔH0" are mentioned? Check the typing of variables in l. 144-149.
Our response: thanks for the suggestion, the text was accordingly modified
-Table 2, 3: reduce font size to have superscripts in same raw. The decimal symbol in English is the dot.
Our response: thanks for the suggestion, the text was accordingly modified
-'L': no need for ', likewise a, b*
Our response: thanks for the suggestion, the text was accordingly modified
-"process, and": never use a comma before "and" or "or" when for parathesis of similar things.
Our response: thanks for the suggestion, the text was accordingly modified
-references: same formatting in all section
Our response: thanks for the suggestion, the text was accordingly modified
You have tried to attribute the colour changes in the crystalline microstructure correctly, what about the influence of HF solution or the etching technique? What is your proposition? What about the human-eye capability of identifying the colour changes (ΔΕ*>5)? What about restorations in front teeth (vertical reflexion) or the back teeth (horizontal replacements)? Please comment and correlate your results where possible.
Our response: thanks for the comment. The HF solution or etching technique is part of the bonding procedure only for lithium disilicate. The same could be supposed to the sandblasting procedure which must be performed on zirconia. However, previous studies showed how HF cannot influence transparency Turgut et al, J Adv Prosthodont 2014) and cannot influence thick ceramics (Turgut et al, J Prosthet Dent 2014). Regarding the vertical vs horizontal reflexions, the study was centered on aesthetic management of frontal teeth only, in which the color matching problem is still high above all when different background shades should be matched.
Reviewer 2 Report
The article is well written and interesting.
I find the subject out of the scope of Polymers, but it may be considered a match to one of the SI's listed subjects, namely "Use of spectrophotometers, photographic equipment, colorimeters to evaluate optical properties and color stability." The decision upon this matter belongs to the Editorial Office.
My suggestions for improving the quality of this article are as follows:
- please use square brackets for citing references
- please use the MDPI style for the reference list
- line 72: "using a standardized specimen thicknesses". Which standard are you referring to? Please correct the language, as well.
- on which basis did you select the experimental protocol?
- to better support your work, please add relevant images of the spectrophotometric assessment, and of the different samples used.
Author Response
My suggestions for improving the quality of this article are as follows:
please use square brackets for citing references
Our response: thanks for the suggestion, the text was accordingly modified
please use the MDPI style for the reference list
Our response: thanks for the suggestion, the text was accordingly modified
line 72: "using a standardized specimen thicknesses". Which standard are you referring to? Please correct the language, as well.
Our response: thanks for the suggestion, the text was accordingly modified
on which basis did you select the experimental protocol?
Our response: thanks for the comment, the experimental protocol was selected according to previous literature. An experimental design section was added to better explain it.
to better support your work, please add relevant images of the spectrophotometric assessment, and of the different samples used.
Our response: thanks for the suggestion, a schematic representation of the sample preparation was added.
Round 2
Reviewer 1 Report
Consider the following:
s instead of seconds, check throughout the text
L*, a*, b* instead of L*a*b*, , check throughout the text
Author Response
Thanks for the comments, the text was accordingly modified.